# Analysis of Metabolites and Metabolic Pathways of Three Chinese Jujube Cultivar

**DOI:** 10.3390/metabo13060714

**Published:** 2023-05-31

**Authors:** Hongxia Liu, Mengyao Yuan, Hui Liu, Lefei Wang, Xusheng Zhao

**Affiliations:** Jujube Scientific Research and Applied Center, Life Science College, Luoyang Normal University, Luoyang 471934, China

**Keywords:** jujube, ultra-high performance liquid chromatography-mass spectrometry (UPLC-MS/MS), untargeted metabolomics, principal component analysis (PCA), hierarchical cluster analysis (HCA)

## Abstract

Most studies on metabolites in jujube fruits focus on specific types of metabolites, but there are only a few comprehensive reports on the metabolites in jujube fruits. In order to understand the variance of metabolites in fruits of different jujube varieties. The objective of this study was to explore the metabolic components of jujube fruit by comparing three cultivars, namely Linyi LiZao (LZ), Jiaocheng SuantianZao (STZ), and Xianxian Muzao (MZ). The metabolites present in the fruits of these three cultivars were evaluated and compared. The results revealed the detection of 1059 metabolites across the three jujube varieties, with each cultivar exhibiting distinct metabolic characteristics. Notably, MZ exhibited a higher abundance of six metabolite classes, namely amino acids and derivatives, flavonoids, lipids, organic acids, phenolic acids, and terpenoids, compared to LZ. Conversely, LZ exhibited higher concentrations of alkaloids, lignans, coumarins, nucleotides, and their derivatives compared to the other two cultivars. In terms of STZ, its content of amino acids and derivatives, lignans and coumarins, organic acids, and phenolic acids was largely similar to that of LZ. However, the content of alkaloids, nucleotides, and their derivatives, and terpenoids was significantly higher in STZ compared to LZ. Additionally, STZ exhibited lower levels of flavonoids and lipids compared to LZ. Moreover, MZ was found to be less nutritionally rich than STZ, except for lignans and coumarins, as it displayed lower levels of all the metabolites. KEGG pathway enrichment analysis revealed six significantly different metabolic pathways (*p* < 0.05) between LZ and MZ, including arginine and proline metabolism, sphingolipid metabolism, flavonoid biosynthesis, glutathione metabolism, glycerophospholipid metabolism, and cysteine and methionine metabolism. The metabolites in STZ and MZ exhibited three significantly different pathways (*p* < 0.05), primarily associated with flavonoid biosynthesis, arginine and proline metabolism, and sphingolipid metabolism. The significantly differential metabolites between LZ and STZ were observed in the phenylpropionic acid biosynthesis pathway and the ubiquinone and other terpenoid-quinone biosynthesis pathways. LZ showed a closer relationship with STZ than with MZ. STZ and LZ exhibited higher medicinal values, while LZ had lower acidity and MZ displayed better antioxidant activity. This study presents the first thorough analysis of metabolites in LZ, STZ, and MZ cultivars, which can serve as a theoretical basis for quality analysis, functional research, and classification processing of jujube fruit.

## 1. Introduction

*Ziziphus jujuba* mill is a plant belonging to the Jujuba genus and is believed to have originated in China more than 8000 years ago. China is the leading producer of jujube globally, both in terms of cultivation area and yield, accounting for over 98% of the total yield [1,2]. Jujube is a traditional fruit with historical significance in both medicine and food. It contains a variety of essential nutrients and medicinal compounds that play a crucial role in traditional Chinese medicine and daily dietary requirements [3,4,5,6]. In modern medicine, jujube has shown beneficial auxiliary effects in treating conditions such as tumors, hypertension, and high cholesterol. It also exhibits several pharmacological properties, including increased blood circulation, enhanced immunity, antioxidant, and anti-aging effects, liver protection, and lipid reduction [7,8,9]. These medicinal properties of jujube are closely associated with its natural active ingredients [10,11].

The jujube varieties LZ, STZ, and MZ are very important for daily consumption for their nutritional value. Wu et al. found that LZ had high levels of ascorbic acids, proanthocyanidins, phenolics, and antioxidant activity [12]. Ye et al. showed that the nutritional quality of LZ was significantly promoted under the action of organic fertilizer [13]. According to Li Zhang’s research, MZ was rich in polysaccharides, which have physicochemical characteristics and bioactivities [14]. However, the relationship between the nutritional and metabolic components of jujube remains unclear. While several studies have investigated jujube metabolites, most of them have focused on specific categories such as sugars [15], amino acids [16,17], and fatty acids [18]. A comprehensive analysis of jujube fruit metabolites, particularly the variations in metabolic components among different jujube varieties, is currently lacking.

Metabolomics involves the qualitative and quantitative analysis of low molecular-weight molecules with a relative molecular mass below 1000 Da. It finds applications in various fields, including food science, disease diagnosis, botany, and toxicology [19,20,21,22,23,24,25]. For instance, Cong et al. conducted a metabolomics study to differentially analyze the composition of peanuts with different colors [26]. Hongrui et al. explored the impact of origin on metabolites in Maca and demonstrated the potential of metabolomics in determining the origin of Maca [27]. Dan et al. investigated the variation of volatile components in sun-grown black tea during storage [28]. Untargeted metabolomic analysis, a novel technology, provides a highly sensitive and accurate method for the simultaneous quantification of thousands of metabolites [29,30].

The objective of this study was to comprehensively understand the relationship between nutritional and metabolic components and compare the variations in metabolites among three major jujube cultivars. To achieve this, we employed an untargeted metabolomic approach using ultra-performance liquid chromatography-mass spectrometry (UPLC-MS/MS) combined with multivariate statistical analysis to measure and compare the metabolites present in the fruits of LZ, STZ, and MZ cultivars. We examined and analyzed the primary metabolic pathways and metabolites in these jujube fruits. The findings of this study contribute to establishing a theoretical foundation and reference for determining the quality, functionality, classification, and processing of jujube fruit.

## 2. Materials and Methods

### 2.1. Samples and Reagents

In August 2020, the fruit (excluding leaves) of *Ziziphus jujuba* Mill cultivars LZ, STZ, and MZ were obtained from the Jujube Experimental Station at Luoyang Normal University in Luoyang, Henan, China. Random selection was made among three jujube trees with consistent phenotypic characteristics, including similar health, age, and height, for the purpose of sample collection. A comprehensive three-dimensional sampling approach was employed for each jujube tree. Following vacuum freeze-drying, the fruits of all three jujube varieties were ground using a mixer mill (MM 400, Retsch) operating at 30 Hz. For sample preparation, 100 mg of powdered material was dissolved in 1.2 mL of 70% methanol. The solution was subjected to vortexing six times, with an interval of 30 min and each vortexing session lasting 30 s. Subsequently, the solution was stored at 4 °C overnight. After centrifugation at 12,000 rpm for 10 min, the supernatant was filtered through a microporous membrane (0.22 µm). The resulting supernatant was collected in an injection flask and subjected to analysis using UPLC-MS/MS. HPLC-grade formic acid, methanol, and acetonitrile were procured from Fisher (Pittsburgh, PA, USA).

### 2.2. UPLC and ESI-Q TRAP-MS/MS Conditions

To analyze the extracts from the samples, a UPLC-ESI-MS/MS instrument (UPLC, SHIMADZU Nexera X2; MS, Applied Biosystems 6500 Q TRAP) was utilized. The analytical conditions were as follows: An Agilent SB-C18 column (1.8 µm, 2.1 mm × 100 mm) was employed during the analysis. The mobile phase consisted of two solvents: solvent A, which was pure water containing 0.1% formic acid, and solvent B, which was acetonitrile with the same concentration of formic acid. A gradient algorithm was applied with initial conditions of 95% A and 5% B for sample readings. Within 9 min, a linear gradient was programmed to transition from 5% A to 95% B and maintained for 1 min. Subsequently, the ratio was adjusted to 95% A and 5% B over 1.1 min and held for 2.9 min. The flow rate was set at 0.35 mL/min, the column oven was maintained at 40 °C, and the injection volume was 2 µL. Finally, the effluent was directed to an ESI-triple quadrupole-linear ion trap (Q TRAP)-MS.

The ESI Turbo Ion-Spray interface-equipped AB6500 Q TRAP UPLC/MS/MS system was utilized for performing Q TRAP and LIT scans. This system had the capability to switch between positive and negative ions, controlled by the Analyst 1.6.3 software (AB Sciex). The ESI source was operated with the following settings: turbo spray ion source, source temperature of 550 °C, ion spray voltage (IS) of 5500 V (positive ion mode)/−4500 V (negative ion mode), and gas pressures of 50, 60, and 25.0 psi for ion source gases I (GSI), gas II (GSII), and curtain gas (CUR), respectively. High collision-activated dissociation (CAD) was employed. To calibrate the instrument and perform mass calibration, solutions of polypropylene glycol with concentrations of 10 mol/L and 100 mol/L were used for the triple quadrupole (QQQ) and linear ion trap (LIT) modes, respectively. Multiple reaction monitoring (MRM) studies were conducted to generate QQQ scans, employing a medium setting for the collision gas (nitrogen). The collision energy (CE) and de-clustering potential (DP) for each MRM transition were optimized, and final values were determined after additional CE and DP optimization. A specific set of MRM transitions was monitored based on the elution times of the metabolites.

### 2.3. Qualitative and Quantitative Analyses of the Constituent Compounds in Jujube Fruit

Metabolite annotations and qualitative assessments were conducted utilizing both a publicly available metabolite database and a proprietary database, MWDB (Metware Biotechnology Co., Ltd., Wuhan, China), drawing on primary and secondary MS data. To ensure the integrity and accuracy of metabolite annotations, we methodically eliminated potential error contributors by removing redundant ion signals such as NH_4_^+^, Na^+^, and K^+^, fragment ions, and isotope signals. Subsequently, we conducted an in-depth analysis of metabolite structures using resources from both MWDB’s in-house database and various publicly accessible mass spectrometry datasets. These include KNAPSAcK (http://kanaya.naist.jp/KNApSAcK/, accessed on 10 March 2022), MassBank (http://www.massbank.jp/, accessed on 11 March 2022), MoToDB (http://www.ab.wur.nl/moto/, accessed on 13 March 2022), HMDB (http://www.hmdb.ca/, accessed on 15 March 2022), PubChem (https://pubchemblog.ncbi.nlm.nih.gov/, accessed on 18 March 2022), ChemBank (http://chembank.med.harvard.edu/compounds, accessed on 18 March 2022), METLIN (http://metlin.scripps.edu/index.php, accessed on 19 March 2022), and NIST Chemistry Webbook (http://webbook.nist.gov/, accessed on 19 March 2022).

The Analyst program was employed for processing the metabolomics data. Metabolite concentrations were determined using the MRM mode of QQQ mass spectrometry. In the MRM mode, interference was eliminated by filtering the precursor ions specific to the target metabolite and removing ions that corresponded to different molecular weights using the quadrupole. To estimate the peak area of the identified compounds in jujube fruits, MultiQuant 3.0.2 (AB Sciex) was used. The peak areas of all identified chemicals were combined for comparative analysis and to assess their relative quantities. The relative amounts of metabolites were then calculated based on the peak areas of the chromatogram.

### 2.4. Untargeted Metabolomics Analysis

Unsupervised principal component analysis (PCA) was performed using the prcomp statistical function in R (www.r-project.org, accessed on 25 March 2022). Prior to conducting unsupervised PCA, the data were normalized to have unit variance. The results of the hierarchical cluster analysis (HCA) for both samples and metabolites were visualized as heat maps with dendrograms. Heat maps were generated to display Pearson’s correlation coefficients (PCC) between samples, computed using the cor function in R. The HCA and PCC analyses were performed using the “Complex Heatmap” package in R. Metabolite signal intensities were normalized by scaling to unit variance and represented as a color spectrum in the heat maps for HCA. Differential regulation of metabolites across groups was determined using a cutoff of VIP (variable importance in projection) ≥1 and an absolute log2FC (fold change) ≥1. Fold-change values were calculated on a linear scale using the least-squares mean [31]. VIP scores were obtained from the data using orthogonal partial least squares discriminant analysis (OPLS-DA). Permutation plots and score plots were also generated to support the findings. The investigation utilized the “Metabolic Analyst R” package. Prior to performing OPLS-DA, the data underwent log transformation (log2) and mean-centering to minimize overfitting. A permutation test with 200 permutations was conducted to assess the robustness of the results. Identified metabolites were annotated using the Kyoto Encyclopedia of Genes and Genomes (KEGG) compound database (http://www.kegg.jp/kegg/compound/, accessed on 25 March 2022) and mapped to the KEGG pathway database (http://www.kegg.jp/kegg/pathway.html, accessed on 25 March 2022). Subsequently, hypergeometric *p*-values were employed for metabolite set enrichment analysis (MSEA) to determine the significance of pathways containing substantially regulated metabolites.

### 2.5. Data Analysis

SPSS (version 3.0, IBM Corporation, Armonk, NY, USA) was used to conduct statistical analysis. The significant differences were determined using Duncan multiple range tests and one-way ANOVA. A *p*-value below 0.05 was marked as significant. OriginPro 2016 (Northampton, MA, USA) was adopted for the figure plotting.

## 3. Results

### 3.1. Metabolism Analysis Based on PCA and HCA Analysis of Jujube Fruit from LZ, STZ, and MZ

Using an untargeted metabolomics approach, we detected a total of 1059 chemical compounds in the jujube fruit across three different cultivars. These compounds were categorized into 11 known classes based on their structural characteristics (Table 1 and Appendix A). Notably, the jujube fruit exhibited a rich composition of metabolites, with flavonoids (195 compounds, 18.41%), lipids (147 compounds, 13.88%), and phenolic acids (136 compounds, 12.84%) being the predominant classes.

Figure 1a presents the coefficient of variation (CV) for samples obtained from the three cultivars and multiple quality control (QC) samples. The higher stability of the experimental data is indicated by a larger proportion of metabolites in the QC samples with lower CV values [32]. Our study demonstrated that over 85% of the metabolites in the QC samples exhibited CV values below 0.3, suggesting a highly reliable and stable experimental dataset.

To analyze the 1059 total metabolites across the three jujube fruit cultivars, we performed PCA (Figure 1b). The results revealed that the first component, PC1, accounted for the major variance among the samples, representing 39.27% of the X variance. In contrast, PC2 accounted for a smaller portion (34% of the variance). Collectively, the first two principal components explained 73.27% of the metabolic variances across all samples, indicating a reliable model. PC1 effectively separated the STZ cultivar from LZ and MZ, while PC2 distinguished LZ and MZ from each other, demonstrating a noticeable trend of separation among the three cultivars. HCA (Figure 1c) revealed that LZ and MZ exhibited similar metabolite types and contents, clustering together, while STZ formed a distinct category separate from LZ and MZ. Both PCA and HCA demonstrated distinct metabolic characteristics among the three jujube cultivars. Wang et al. combined PCA and HCA to detect compounds in 15 cultivars of jujube fruits, which indicated that 15 cultivars were categorized into 6 groups [33].

### 3.2. Differential Metabolite Screening in the Jujube Fruit of LZ, STZ, and MZ

The metabolite differences were assessed based on VIP scores >1 and FC values ≥2 or ≤0.5 in each group sample. FC, which represents the fold change between experimental and control groups, is used to determine statistically significant differences. An FC of ≥2 indicates upregulation, while an FC of ≤ 0.5 indicates downregulation. The VIP value indicates the degree of differentiation between associated metabolites in different groups, with particular significance when the VIP ≥ 1 [34]. A volcano plot, displaying VIP and fold change values (Appendix A), was used to identify differential metabolites. Each point in the plot represents a different metabolite, and the *x*-axis represents the fold change value (log2) for each substance within the group.

The screening results for differential metabolites are shown in Appendix A. In total, 102 metabolites were upregulated, 198 metabolites were downregulated, and the others changed insignificantly in MZ compared to those in LZ (Appendix A). Comparing LZ and STZ (Appendix A), 155 metabolites were upregulated, 143 metabolites were downregulated, and the others changed insignificantly. MZ metabolites differ from STZ metabolites in 323 metabolites, with 107 upregulated and 216 downregulated metabolites, and the others changed insignificantly (Appendix A). Yan screened differential metabolites between Chinese and North American wild rice. There were 160 upregulated and 197 downregulated metabolites and 315 metabolites showed no significant difference between the two groups [35].

A Venn diagram analysis was conducted to identify common and unique metabolites among the various comparison groups. The comparison of LZ vs. MZ, LZ vs. STZ, and STZ vs. MZ revealed 59 common differential metabolites among the three groups, as shown in Appendix A. LZ vs. STZ and STZ vs. MZ had 57 unique differential metabolites each, while LZ vs. MZ had 54 unique differential metabolites.

To gain a better understanding of the changes in jujube fruit across the three cultivars, the significantly different metabolites were categorized into nine major classes, and a comparison was made. It was observed that MZ fruits exhibited decreased metabolite ion intensity compared to LZ fruits. Among the nine classes of metabolites, including amino acids and derivatives, flavonoids, lipids, organic acids, phenolic acids, and terpenoids, six were found to be less abundant in MZ than LZ, whereas alkaloids, lignans, and coumarins, and nucleotides and their derivatives were more concentrated in LZ (Figure 2a). When comparing LZ and STZ, the content of amino acids and derivatives, lignans and coumarins, organic acids, and phenolic acids was nearly equal, while the content of alkaloids, nucleotides, and their derivatives, as well as terpenoids, was significantly higher in STZ compared to LZ. Moreover, the contents of flavonoids and lipids were lower in STZ compared to LZ (Figure 2b). It was evident that MZ fruits had lower nutritional value than STZ fruits, as indicated by lower metabolite ion intensity in all metabolites except for lignans and coumarins (Figure 2c).

### 3.3. Analysis of Unique Differential Metabolites

A total of 54 unique differential metabolites were identified in the comparison between LZ and MZ. These included 15 flavonoids, eight lipids, seven terpenoids, five phenolic acids, five alkaloids, and four nucleotides and their derivatives (Appendix A). Among these metabolites, 13 were significantly upregulated in MZ and exhibited 2-fold higher levels (2.17 < FC < 2.72) compared to LZ. Additionally, 41 metabolites were significantly downregulated in MZ, with their content being less than half (FC < 0.5) of that in LZ.

In the comparison between LZ and STZ, 57 unique differential metabolites were identified. Appendix A presents the main differences in metabolites between LZ and STZ, including nine alkaloids, seven flavones, seven lipids, six phenolic acids, six terpenoids, five amino acids, and their derivatives, five nucleotides and their derivatives, three organic acids, one lignan and coumarin, one tannin, and seven others. Among these metabolites, 26 were significantly upregulated in STZ, exhibiting levels 2 times higher (2 < FC < 14.43) than in LZ. Notably, N-Acetyl-L-tyrosine showed a remarkable upregulation in STZ, being 14.43 times higher than in LZ. Furthermore, 31 metabolites were significantly downregulated in MZ, with levels less than 50% (FC < 0.5) of those in LZ.

In the comparison between STZ and MZ, there were 57 unique differential metabolites (Appendix A), including 19 flavonoids, 10 lipids, eight amino acids, and their derivatives, four phenolic acids, four terpenoids, four organic acids, two tannins, two alkaloids, one nucleotide and its derivatives, one lignan and coumarins, and two others. Among these metabolites, 19 were significantly upregulated in MZ, showing levels 2 times higher (2 < FC < 3.87) than in STZ. On the other hand, 38 metabolites were significantly downregulated in MZ, with levels less than half (FC < 0.5) of those in STZ. The metabolite with the lowest content in MZ was N-Acetyl-L-Arginine, which was only 0.07 times that in STZ.

### 3.4. Differential Metabolite Pathway Analysis

The metabolic pathways in which the differential metabolites were involved were analyzed using KEGG pathway enrichment. Figure 3 presents the results of the KEGG pathway enrichment analysis for the differential metabolites in the three jujube fruit cultivars. The metabolites that differed between LZ and MZ were distributed across 75 metabolic pathways, with six pathways showing significant differences (*p* < 0.05) (Figure 3A). These pathways included arginine and proline metabolism, sphingolipid metabolism, flavonoid biosynthesis, glutathione metabolism, cysteine and methionine metabolism, and glycerophospholipid metabolism. Various components participate in the arginine and proline metabolic pathways, such as four alkaloids, four amino acids and their derivatives, and two organic acids. Among these, the synthesis of agmatine (FC = 4.12), putrescine (FC = 9.45), N-acetylputrescine (FC = 3.86), spermidine (FC = 5.99), S-(5’-adenosine)-L-methionine (FC = 2.29), and pyruvate (FC = 34,832.59) was more active and accumulated in MZ. Conversely, 4-guanidine butyric acid, L-ornithine, trans-4-hydroxy-L-proline, and L-glutamic acid exhibited higher metabolic activity in LZ (with FC < 0.5). The metabolism of sphingolipids involved four different metabolites (one alkaloid, one amino acid and its derivatives, and two lipids), including phosphoethanolamine (FC = 0.34), 3-dehydrodihydrosphingolipin (FC = 0.17), 4-hydroxysphingolipin (FC = 0.12), and L-serine (FC = 0.49), which also showed higher metabolic activity in LZ (with FC < 0.5). In the flavonoid biosynthesis pathway, 13 different metabolites (one phenolic acid and 12 flavonoids) were involved. In LZ, the synthesis of flavonoid compounds such as garbanzol, pinobanksin, 2’,4,4’,6’-tetrahydroxychalcone, luteolin, aromadendrin, fustin, quercetin, 5-hydroxyflavone, taxifolin, prunin, and neohesperidin was more active (FC < 0.5). However, 5-O-p-coumadyl quinic acid (FC = 4.47) and naringin (FC = 67.89) were synthesized and metabolized to a greater extent in MZ.

The metabolites that exhibited variations between LZ and STZ were found to be involved in a total of 79 metabolic pathways. Notably, two of these pathways displayed significant differences (with a *p*-value < 0.05), as illustrated in Figure 3B. These pathways encompass phenylpropanoid biosynthesis, ubiquinone biosynthesis, and terpenoid-quinone biosynthesis. Among the nine distinct metabolites investigated, there were an amino acid and its derivatives, lignan and coumarin, as well as seven phenolic acids. LZ demonstrated higher metabolic activity for cinnamic acid, caffeic acid, Scopoletin (7-Hydroxy-6-methoxycoumarin), sinapic acid, coniferin, chlorogenic acid, syringin, and 1-O-Sinapoyl-D-glucose (fold change < 0.5). On the other hand, L-Tyrosine (fold change = 2.32), Sinapyl alcohol (fold change = 4.13), and 5-O-p-Coumaroylquinic acid (fold change = 9.59) exhibited greater activity and accumulated in STZ.

The metabolites that exhibited differential expression between STZ and MZ were found to participate in a total of 74 metabolic pathways. Notably, three of these pathways displayed significant differences (with a *p*-value < 0.05) (Figure 3C). These pathways include flavonoid biosynthesis, arginine and proline metabolism, and sphingolipid metabolism, which align with the significant differences observed in the pathways between LZ and MZ. The flavonoid biosynthesis pathway comprised 12 distinct metabolites, including two phenolic acids and 10 flavonoids. Among them, pinobanksin, naringenin chalcone, 2′,4,4′,6′-tetrahydroxychalcone, phloretin, eriodictyol (5,7,3′,4′-tetrahydroxyflavanone), quercetin, epigallocatechin, prunin, phlorizin, naringin, and neohesperidin exhibited higher levels in STZ (FC < 0.5). Only chlorogenic acid in STZ was found to be lower, comprising less than 1% of MZ. The metabolic pathway of arginine and proline involved eight different metabolites, including five alkaloids, two amino acids and their derivatives, and one organic acid. The concentrations of putrescine (FC = 22.45), N-acetylputrescine (FC = 3.48), agmatine (FC = 3.58), and spermidine (FC = 2.71) were higher in MZ. STZ exhibited higher levels of L-ornithol, L-glutamic acid, 4-guanidine butyric acid, and p-coumaroylagmatine. The sphingolipid metabolism pathway involved three distinct metabolites, including 3-dehydrosphinganine, o-phosphorylethanolamine, and 4-hydroxysphinganine. Notably, o-phosphorylethanolamine (FC = 0.43) and 4-hydroxysphinganine (FC = 0.33) were more easily metabolized and synthesized in STZ. MZ exhibited a higher concentration of 3-dehydrosphinganine (FC = 8.95).

## 4. Discussion

In recent years, many studies have investigated the metabolites of jujube fruit. Feng et al. detected 40 metabolites in different developmental stages of jujube based on UHPLC-MS technology [36]. Zhang et al. detected 705 compounds grouped into 23 classes using HPLC with a UV detector [37], and Shi et al. recorded the changing levels of phenolic compounds, anthocyanins, carotenoids, and chlorophylls during fruit development [38]. In this study, untargeted metabolomics analysis using UHPLC-MS/MS identified 1059 metabolites from 11 different classes in LZ, STZ, and MZ fruits. These included 195 distinct flavonoids, 147 different lipids, 136 different phenolic acids, 112 different terpenoids, 95 different amino acids and their derivatives, 86 different alkaloids, 83 different organic acids, 55 different nucleotides, and their derivatives, 26 different lignans and coumarins, and 13 different tannins, which constituted the most abundant primary metabolites. The metabolite diversity and abundance surpassed those of previous studies.

There were 54 unique differential metabolites of LZ vs. MZ. Compared with the fruits of LZ, a decrease was observed in the metabolites of MZ, 41 of which were significantly lower in MZ, including six primary metabolite classes—amino acids and derivatives, flavonoids, lipids, organic acids, phenolic acids, and terpenoids—that were found to be lower in MZ compared to LZ. On the other hand, LZ exhibited a higher abundance of alkaloids, lignans, coumarins, nucleotides, and their derivatives. In previous studies, the incompatible soybean variety PI437654 and three compatible soybean varieties were used to clarify the differences in metabolites, and 14 unique differential metabolites were found [39]. Lignans, which are non-flavonoid polyphenols composed of two phenylpropyl derivatives, possess pharmacological properties such as antitumor and antioxidant effects. They can bind to estrogen receptors and exhibit phytoestrogenic activity, suggesting their potential for therapeutic development [40]. Analyzing the ion intensity of metabolites between LZ and STZ revealed that STZ had a higher content of alkaloids, nucleotides, and their derivatives, and terpenoids. Conversely, the content of flavonoids and lipids was lower in STZ compared to LZ. Moreover, LZ exhibited a higher abundance of alkaloids, nucleotides, and their derivatives compared to MZ and STZ. Alkaloids are nitrogenous organic compounds with alkaline properties and complex ring structures known for their significant biological activity [41]. Nucleotides are compounds comprising a purine or pyrimidine base, ribose or deoxyribose, and phosphoric acid. They play essential roles in numerous biological functions [42]. The ion intensity of all metabolites, except lignans and coumarins, was lower in MZ fruits compared to STZ fruits, indicating a lower nutritional content in the former. 

In comparison to LZ, the majority of metabolites in MZ displayed lower levels among the 54 unique differential metabolites, with only 13 metabolites showing higher content in MZ (Appendix A). These elevated metabolites primarily consisted of lipids, along with a few phenolic acids and flavonoids. Lipids serve various roles related to energy, signaling, and structural functions in maintaining health [11]. Phenolic acids, which include monohydroxybenzoic acids (such as parabens) and dihydroxybenzoic acids (such as gencholic acid and protocatechuic acid), possess activities such as antioxidant, free radical scavenging, anti-ultraviolet radiation, antibacterial, and antiviral effects [43,44]. Flavonoids, a type of polymer, are renowned for their potent antioxidant properties and exhibit benefits such as anticancer, antitumor, hypoglycemic, and hypolipidemic effects [45,46]. Compared to LZ, MZ offers higher energy provision and improved antioxidant activity.

Between LZ and STZ (Appendix A), 26 metabolites were significantly upregulated in STZ. STZ exhibited significant upregulation of Procyanidin A6 (FC = 3.01), 2-Phenylpropionic Acid (FC = 3.26), Tormentic Acid (FC = 3.33), LysoPC 16:0 (2n isomer) (FC = 3.47), and N-Acetyl-L-tyrosine (FC = 14.43). Notably, N-Acetyl-L-tyrosine displayed markedly higher levels in STZ, surpassing LZ by more than 14 times. N-Acetyl-L-tyrosine is an important organic chemical intermediate widely used in medicine, pesticides, and the chemical industry [47], highlighting the higher medicinal value of STZ. When compared to MZ, STZ exhibited 57 unique differential metabolites, with 38 of them having higher levels in STZ. The contents of N-Acetyl-L-Arginine, N-Acetyl-L-Aspartic Acid, LysoPC 20:1, Quercetin-3-O-arabinoside (Guaijaverin), and Jujuboside B1 were significantly higher in STZ than in MZ, ranging from 3 to 18 times higher, indicating a richer nutritional value and greater biological activity in STZ. 

The two pathways, phenylpropanoid biosynthesis and ubiquinone and terpenoid-quinone biosynthesis were found to be significantly different between LZ and STZ. The different metabolites in LZ vs. MZ were distributed across 75 metabolic pathways, six of which were significantly different. The differential metabolites observed in STZ vs. MZ were found to be involved in 74 metabolic pathways, with three of them showing significant differences. Li used the same approach to study the pathways in four buckwheat flowers with different colors. Results showed that the differential metabolites of the comparison groups were mainly distributed to the biosynthesis of isoflavonoids, flavonoids, flavones, and flavonols, secondary metabolites, phenylpropanoids, anthocyanins, and metabolic pathways [48,49]. The arginine and proline metabolic pathways involved eight distinct metabolites, including five alkaloids, two amino acids and their derivatives, and one organic acid. Among these, putrescine (FC = 22.45), N-Acetylputrescine (FC = 3.48), agmatine (FC = 3.58), and spermidine (FC = 2.71) were the most abundant in STZ. Additionally, there were differences observed in sphingolipids when compared to MZ. The sphingolipid metabolism included three metabolic components, namely 3-Dehydrosphinganine, O-Phosphorylethanolamine, and 4-Hydroxysphinganine. O-Phosphorylethanolamine (FC = 0.43) and 4-Hydroxysphinganine (FC = 0.33) were the two metabolites that were most readily metabolized and synthesized in STZ. MZ displayed a higher concentration of 3-Dehydrosphinganine (FC = 8.95).

## 5. Conclusions 

In this research, we employed UPLC-MS/MS and multivariate statistical analysis to explore the relationship between nutritional and metabolic components in three major jujube cultivars: LZ, STZ, and MZ. The results revealed that 1059 metabolites were detected among the three jujube varieties. PCA and HCA indicated that each cultivar had distinct metabolic properties. Differential metabolite screening in the jujube fruit of LZ, STZ, and MZ showed that 102 metabolites were upregulated and 198 metabolites were downregulated in MZ compared with the metabolites in LZ. Comparing LZ and STZ, 155 metabolites were upregulated and 143 metabolites were downregulated. MZ metabolites differ from STZ metabolites in 323 metabolites, with 107 upregulated and 216 downregulated. KEGG pathway enrichment analysis highlighted that the different metabolites in LZ vs. MZ were distributed across 75 metabolic pathways, with six pathways exhibiting significant differences (*p* < 0.05). These pathways encompassed arginine and proline metabolism, sphingolipid metabolism, flavonoid biosynthesis, glutathione metabolism, cysteine and methionine metabolism, and glycerophospholipid metabolism. Furthermore, phenylpropanoid biosynthesis and ubiquinone and terpenoid-quinone biosynthesis were identified as the significantly differing metabolites between LZ and STZ. On the other hand, the differential metabolites in STZ vs. MZ were involved in 74 metabolic pathways, with three pathways showing significant differences (*p* < 0.05) (Figure 3C), namely flavonoid biosynthesis, arginine and proline metabolism, and sphingolipid metabolism.

Our results indicated that MZ exhibited closer metabolic similarities to LZ, whereas STZ displayed closer metabolic similarities to MZ. The differences in metabolites between LZ and STZ, as well as between LZ and MZ, were relatively small, while the disparity between STZ and MZ was significant. STZ exhibited higher medicinal value, enhanced flavonoid biosynthesis, and a greater synthesis of flavonoids. The phenylpropionic acid biosynthesis pathway in LZ demonstrated greater activity, leading to the accumulation of phenolic acids with higher medicinal value. Additionally, the higher content of anthocyanins in MZ contributed to its superior antioxidant activity. 

## Figures and Tables

**Figure 1 metabolites-13-00714-f001:**
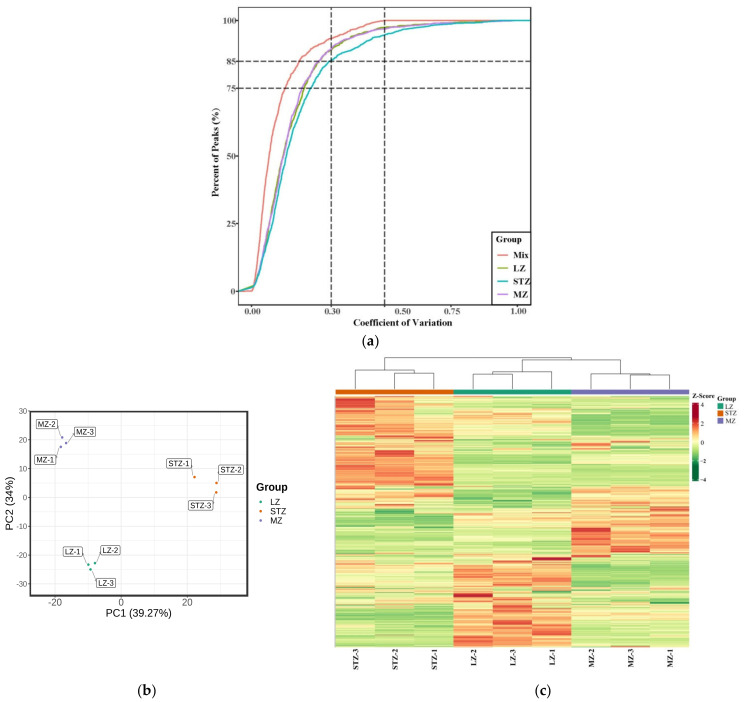
(**a**) Plot of the coefficients of variation (CV) for different cultivars and multiple QC samples. (**b**) A scatter plot showing PCA scores for all metabolites; (**c**) HCA of jujube fruit from LZ, STZ, and MZ. Note: high metabolite levels are shown in red; low levels are shown in green.QC: quality control; PCA: principal component analysis; HCA: hierarchical cluster analysis; LZ: Linyi LiZao; STZ: Jiaocheng SuantianZao; MZ: Xianxian Muzao.

**Figure 2 metabolites-13-00714-f002:**
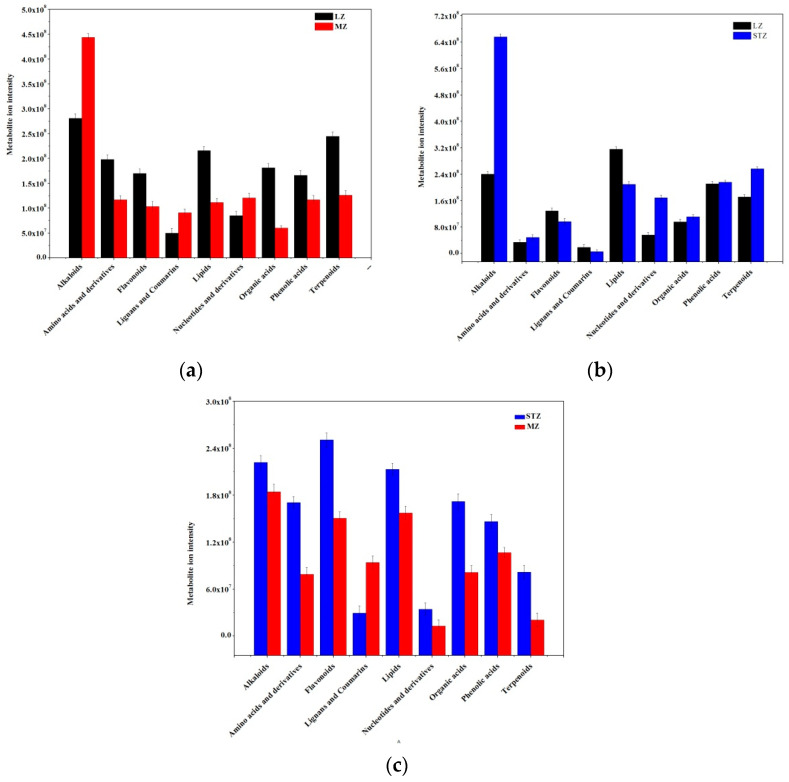
Comparison of the total ion intensity of various classes of metabolites among three Chinese jujube cultivars: (**a**) LZ vs. MZ; (**b**) LZ vs. STZ; (**c**) STZ vs. MZ. LZ: Linyi LiZao; STZ: Jiaocheng SuantianZao; MZ: Xianxian Muzao.

**Figure 3 metabolites-13-00714-f003:**
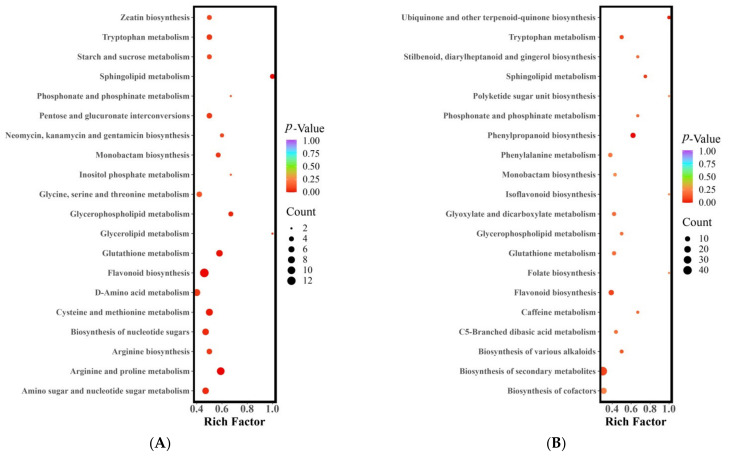
KEGG enrichment of differential metabolites (**A**) LZ vs. MZ, (**B**) LZ vs. STZ, and (**C**) STZ vs. MZ. Abscissa, the Rich Factor correlates to each pathway; ordinate, the name of the pathways; and color, the *p*-value. The darker the hue, the higher the richness. The size of the dot indicates the number of differentially enriched metabolites. LZ: Linyi LiZao; STZ: Jiaocheng SuantianZao; MZ: Xianxian Muzao.

**Table 1 metabolites-13-00714-t001:** Classification of metabolites in the jujube fruit of STZ, LZ, and MZ.

Class	Number of Compounds	Class	Number of Compounds
Alkaloids	86	Organic acids	83
Amino acids and derivatives	95	Phenolic acids	136
Flavonoids	195	Tannins	13
Lignans and Coumarins	26	Others	111
Lipids	147		

Others include ketone, saccharides, Vitamins, aldehydes, and Lactones.

## Data Availability

Data will be made available on request.

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
