# Peer review of "Analysis of Metabolites and Metabolic Pathways of Three Chinese Jujube Cultivar"

_metabolites, 2023, doi:10.3390/metabo13060714_

Round 1

Reviewer 1 Report (Previous Reviewer 1)

The present research work compared the metabolites and metabolic pathways of three cultivars of Chinese Jujube fruits using LC-MS methodology. Regrettably, this study does not exhibit novelty in its findings, and suffers from a dearth of well-executed and well-designed experimental protocols. Furthermore, the manuscript's readability is wanting. Taken together, the quality of this study is insufficient for publication.

1.      The English language of this manuscript requires significant improvement.

2.      The analytical method's reliability and stability must be validated by monitoring multiple QC samples.

3.      It appears that Figure 3 is absent from the manuscript.

4.      The discussion section necessitates a complete rewrite, as most of its contents do not deliberate or reflect on the findings.

5.      The title of section 2.5 is incorrect.

6.      The term "widely targeted" is not widely utilized, and should be replaced with "untargeted".

7.      In line 131, did the experiment include leaves?

8.      In line 156, what is meant by "unique chemical compounds"?

9.      In lines 194-195, what is the methodology for calculating FC values in "each sample"?

10.  Figure 2 does not appear to be a volcano plot.

Author Response

Dear editor and reviewers:

Firstly, we would like to express our gratitude for your valuable feedback and acknowledgment of our manuscript, entitled "Analysis of Metabolites and Metabolic Pathways of Three Chinese Jujube Cultivars" (metabolites-2117342), as well as your kind assistance in improving our manuscript. On the other hand, in accordance with the reviewers' comments, there are indeed several shortcomings in this manuscript, including typographical errors, grammatical mistakes, and deficiencies in the related content. These observations are invaluable and extremely beneficial for enhancing and refining our paper, as well as providing significant guidance for our future research endeavors. We have diligently reviewed the comments and made the appropriate corrections, which we hope meet your approval. For your convenience in verifying the changes, the revisions are highlighted in red within the revised manuscript. We trust that the modifications are satisfactory and eagerly await your response. The adjustments made based on the reviewers' feedback are detailed below:

Reviewers' comments:

Reviewer #1: The present research work compared the metabolites and metabolic pathways of three cultivars of Chinese Jujube fruits using LC-MS methodology. Regrettably, this study does not exhibit novelty in its findings, and suffers from a dearth of well-executed and well-designed experimental protocols. Furthermore, the manuscript's readability is wanting. Taken together, the quality of this study is insufficient for publication.

Responses: We have meticulously examined the manuscript, and substantial revisions have been made to the abstract, conclusion, and discussion sections. We trust that these improvements align with your expectations and receive your endorsement.

  1. The English language of this manuscript requires significant improvement.

Responses: We have carefully reviewed the manuscript, and the English language has been significantly enhanced throughout.

  1. The analytical method's reliability and stability must be validated by monitoring multiple QC samples.

Responses: We have incorporated multiple quality control (QC) samples to bolster the reliability and stability of the analytical method in Figure 1a, with the associated data analysis emphasized in red.

  1. It appears that Figure 3 is absent from the manuscript.

Responses: We sincerely apologize for the error we made and have rectified the order of the images.

  1. The discussion section necessitates a complete rewrite, as most of its contents do not deliberate or reflect on the findings.

Responses: The discussion has undergone a complete rewrite. If you have any further suggestions, please do not hesitate to contact us.

  1. The title of Section 2.5 is incorrect.

Responses: We have implemented the change.

  1. The term "widely targeted" is not widely utilized, and should be replaced with "untargeted".

Responses: We have modified the text by replacing the term "widely targeted" with "untargeted".

  1. In line 131, did the experiment include leaves?

Responses: In Section 2.1, we have explicitly stated that leaves were not included in the experiment.

  1. In line 156, what is meant by "unique chemical compounds"?

Responses: We apologize for the error. The word "unique" has been removed.

  1. In lines 194–195, what is the methodology for calculating FC values in "each sample"?

Responses: In Section 2.1, we have included the methodology for calculating FC values. We would like to clarify that the term "each sample" refers to each sample in the group, which has been explicitly stated.

  1. Figure 2 does not appear to be a volcano plot.

Responses: We have updated the relevant section to indicate that the volcano plot can be found in Figure S1A-C.

Reviewer 2 Report (New Reviewer)

The manuscript entitled „Analysis of Metabolites and Metabolic Pathways of Three Chinese Jujube Cultivars” describes a study of metabolic profiles of jujube fruits belonging to different cultivars.

I see that the manuscript has been already improved at the previous stage. Therefore, I have just some additional comments.

Line 52 – you cannot write that “Metabolomics is the study of …”. Please rephrase the sentence.

Line 59 – some problem with reference.

Please explain all the abbreviations in the first use.

Section 2.4. – what do you mean by quantitative analysis? To calculate the content of compounds (not concentration, concentration refers to liquid, not solid fruit) you need the calibration curve of each compound. Did you describe some way of relative calculation? I don’t understand what you mean by comparison to leaves.

In the aim and the abstract, there is information about changes in metabolic profiles affecting flavour, however, no flavour-related studies were performed. Therefore, I suggest to remove the flavour from the text.

The text contains many typos and the capital letters are present with no point. Please rearrange the text.

The conclusions totally don’t respond to what is written in the aim of the study, especially this part “We conducted this study to comprehensively understand the relationship between flavour and nutritional and metabolic components”.

The reference list contains mainly the studies published by the researchers from the country of the authors. Please make the list “more international” and look broader.

Author Response

Dear editor and reviewers:

Firstly, we would like to express our gratitude for your valuable feedback and acknowledgment of our manuscript, entitled "Analysis of Metabolites and Metabolic Pathways of Three Chinese Jujube Cultivars" (metabolites-2117342), as well as your kind assistance in improving our manuscript. On the other hand, in accordance with the reviewers' comments, there are indeed several shortcomings in this manuscript, including typographical errors, grammatical mistakes, and deficiencies in the related content. These observations are invaluable and extremely beneficial for enhancing and refining our paper, as well as providing significant guidance for our future research endeavors. We have diligently reviewed the comments and made the appropriate corrections, which we hope meet your approval. For your convenience in verifying the changes, the revisions are highlighted in red within the revised manuscript. We trust that the modifications are satisfactory and eagerly await your response. The adjustments made based on the reviewers' feedback are detailed below:

Reviewers' comments:

Reviewer #2: The manuscript entitled „Analysis of Metabolites and Metabolic Pathways of Three Chinese Jujube Cultivars” describes a study of metabolic profiles of jujube fruits belonging to different cultivars. I see that the manuscript has been already improved at the previous stage. Therefore, I have just some additional comments.

  1. Line 52 – you cannot write that “Metabolomics is the study of …”. Please rephrase the sentence.

Responses: We appreciate your advice and have rewritten the sentence accordingly.

  1. Line 59 – some problem with reference.

Responses: We appreciate your advice and have implemented the change.

  1. Please explain all the abbreviations in the first use.

Responses: We have carefully reviewed the manuscript and have defined all abbreviations in their first use.

  1. Section 2.4. – what do you mean by quantitative analysis? To calculate the content of compounds (not concentration, concentration refers to liquid, not solid fruit) you need the calibration curve of each compound. Did you describe some way of relative calculation? I don’t understand what you mean by comparison to leaves.

Responses: We have clarified that the relative quantities of metabolites were determined by analyzing the peak areas of the chromatogram. Additionally, we would like to specify that we used fruit instead of leaves in the study.

  1. In the aim and the abstract, there is information about changes in metabolic profiles affecting flavor, however, no flavor-related studies were performed. Therefore, I suggest to remove the flavor from the text.

Responses: We appreciate your valuable feedback, and we have removed the reference to flavor from the text.

  1. The text contains many typos and the capital letters are present with no point. Please rearrange the text.

Responses: Thank you for your valuable advice.

We have checked the manuscript thoroughly and rearranged the text.

  1. The conclusions totally don’t respond to what is written in the aim of the study, especially this part “We conducted this study to comprehensively understand the relationship between flavor and nutritional and metabolic components”.

Responses: We appreciate your helpful advice. Based on it, we have revised our conclusions.

  1. The reference list contains mainly the studies published by the researchers from the country of the authors. Please make the list “more international” and look broader.

Responses: We appreciate your advice. In response, we have incorporated additional international references to broaden the scope of our work.

Reviewer 3 Report (New Reviewer)

Manuscript Title: Analysis of Metabolites and Metabolic Pathways of Three Chinese Jujube Cultivars

Journal: Metabolites

Manuscript ID: metabolites-2278311-peer-

Reviewer Recommendation: Major Revision

Reviewer's comments to Authors

Abstract

·        The abstract should be the overview of the whole manuscript. Thus it should involve the experiment's main finding data, which was missing in the manuscript.

·        The future perspective of the experiment should be mentioned in the abstract.

Novelty

·         The authors failed to give satisfactory answer to the novelty statement. Do revise it

Materials and methods

·        There are some error in the sentence formation so rephrase the whole sentence and avoid using long paragraph in the materials and methods section.

·        There are some grammatical errors, so please revise this section carefully to remove any possible grammatical and typos errors.

References:

The below listed relevant references should be considered and cited appropriately in introduction, material and methods, results, and discussion sections of this manuscript, which will certainly improve the quality of the manuscript significantly.

·         Kumar, D., et al., (2022). Photosynthesis, lipid peroxidation, and antioxidative responses of Helianthus annuus L. against chromium (VI) accumulation. International Journal of Phytoremediation24(6), 590-599.

·         Yadav, M., et al., (2022). Foliar application of α-lipoic acid attenuates cadmium toxicity on photosynthetic pigments and nitrogen metabolism in Solanumlycopersicum L. ActaPhysiologiaePlantarum44(11), 112.

Conclusion

·         Please elaborate the conclusion part as it is very briefly described.

·         Conclusion section failed to enlighten the spirit of the finding. Revise it precisely.

Author Response

Dear editor and reviewers:

Firstly, we would like to express our gratitude for your valuable feedback and acknowledgment of our manuscript, entitled "Analysis of Metabolites and Metabolic Pathways of Three Chinese Jujube Cultivars" (metabolites-2117342), as well as your kind assistance in improving our manuscript. On the other hand, in accordance with the reviewers' comments, there are indeed several shortcomings in this manuscript, including typographical errors, grammatical mistakes, and deficiencies in the related content. These observations are invaluable and extremely beneficial for enhancing and refining our paper, as well as providing significant guidance for our future research endeavors. We have diligently reviewed the comments and made the appropriate corrections, which we hope meet your approval. For your convenience in verifying the changes, the revisions are highlighted in red within the revised manuscript. We trust that the modifications are satisfactory and eagerly await your response. The adjustments made based on the reviewers' feedback are detailed below:

Reviewers' comments:

Reviewer #3 

1.Abstract

The abstract should be the overview of the whole manuscript. Thus it should involve the experiment's main finding data, which was missing in the manuscript.

The future perspective of the experiment should be mentioned in the abstract.

Responses: We have eliminated extraneous content from the manuscript and included the primary experimental findings in the abstract. Furthermore, we have provided a perspective on future research directions in the final section of the abstract.

2.Novelty

The authors failed to give satisfactory answer to the novelty statement. Do revise it

Responses: We have revised the abstract, discussion, and conclusion sections of the manuscript and included a statement regarding the novelty of our findings.

  1. Materials and methods

There are some error in the sentence formation so rephrase the whole sentence and avoid using long paragraph in the materials and methods section.

There are some grammatical errors, so please revise this section carefully to remove any possible grammatical and typos errors.

Responses: We have rephrased this section of the manuscript and hope that it now meets with your approval.

  1. References:

The below listed relevant references should be considered and cited appropriately in introduction, material and methods, results, and discussion sections of this manuscript, which will certainly improve the quality of the manuscript significantly.

Kumar, D., et al., (2022). Photosynthesis, lipid peroxidation, and antioxidative responses of Helianthus annuus L. against chromium (VI) accumulation. International Journal of Phytoremediation, 24(6), 590–599.

Yadav, M., et al., (2022). Foliar application of α-lipoic acid attenuates cadmium toxicity on photosynthetic pigments and nitrogen metabolism in Solanumlycopersicum L. ActaPhysiologiaePlantarum, 44(11), 112.

Responses: We have included two additional references in the introduction section of the manuscript.

  1. Conclusion

Please elaborate the conclusion part as it is very briefly described.

Conclusion section failed to enlighten the spirit of the finding. Revise it precisely.

Responses: We have rephrased the conclusion section of the manuscript and hope that it now meets with your approval.

Round 2

Reviewer 1 Report (Previous Reviewer 1)

The authors well addressed my comments.

Author Response

Response to comments

Dear editor and reviewers:

Firstly, we would like to express our gratitude for your valuable feedback and acknowledgment of our manuscript, entitled "Analysis of Metabolites and Metabolic Pathways of Three Chinese Jujube Cultivars" (metabolites-2117342), as well as your kind assistance in improving our manuscript. On the other hand, in accordance with the reviewers' comments, there are indeed several shortcomings in this manuscript, including typographical errors, grammatical mistakes, and deficiencies in the related content. These observations are invaluable and extremely beneficial for enhancing and refining our paper, as well as providing significant guidance for our future research endeavors. We have diligently reviewed the comments and made the appropriate corrections, which we hope meet your approval. For your convenience in verifying the changes, the revisions are highlighted in red within the revised manuscript. We trust that the modifications are satisfactory and eagerly await your response. The adjustments made based on the reviewers' feedback are detailed below:

Reviewers' comments:

Reviewer #1: The present research work compared the metabolites and metabolic pathways of three cultivars of Chinese Jujube fruits using LC-MS methodology. Regrettably, this study does not exhibit novelty in its findings, and suffers from a dearth of well-executed and well-designed experimental protocols. Furthermore, the manuscript's readability is wanting. Taken together, the quality of this study is insufficient for publication.

Responses: We have meticulously examined the manuscript, and substantial revisions have been made to the abstract, conclusion, and discussion sections. We trust that these improvements align with your expectations and receive your endorsement.

  1. The English language of this manuscript requires significant improvement.

Responses: We have carefully reviewed the manuscript, and the English language has been significantly enhanced throughout.

  1. The analytical method's reliability and stability must be validated by monitoring multiple QC samples.

Responses: We have incorporated multiple quality control (QC) samples to bolster the reliability and stability of the analytical method in Figure 1a, with the associated data analysis emphasized in red.

  1. It appears that Figure 3 is absent from the manuscript.

Responses: We sincerely apologize for the error we made and have rectified the order of the images.

  1. The discussion section necessitates a complete rewrite, as most of its contents do not deliberate or reflect on the findings.

Responses: The discussion has undergone a complete rewrite. If you have any further suggestions, please do not hesitate to contact us.

  1. The title of Section 2.5 is incorrect.

Responses: We have implemented the change.

  1. The term "widely targeted" is not widely utilized, and should be replaced with "untargeted".

Responses: We have modified the text by replacing the term "widely targeted" with "untargeted".

  1. In line 131, did the experiment include leaves?

Responses: In Section 2.1, we have explicitly stated that leaves were not included in the experiment.

  1. In line 156, what is meant by "unique chemical compounds"?

Responses: We apologize for the error. The word "unique" has been removed.

  1. In lines 194–195, what is the methodology for calculating FC values in "each sample"?

Responses: In Section 2.1, we have included the methodology for calculating FC values. We would like to clarify that the term "each sample" refers to each sample in the group, which has been explicitly stated.

  1. Figure 2 does not appear to be a volcano plot.

Responses: We have updated the relevant section to indicate that the volcano plot can be found in Figure S1A-C.

Reviewer 3 Report (New Reviewer)

Still some points need to be addressed before final decision to accept. These are given below

Reviewer Recommendation: Minor Revision

 Reviewer’s comments to Authors

1.      Grammatical errors are present, please revise the whole manuscript to remove any possible grammatical and typos errors.

2.      Error in sentence formation, please revise the whole manuscript to avoid the use of long sentences.

3.      In the keywords, it is strongly advisable to use suitable words that can aid in finding out the manuscript in current registers or indexes. Strictly avoid the use of title words in the keywords.

4.      Please maintain uniformity while in-text citation and referencing.

5.      The reference does not meet the format requirements of the Journal so please check the references as per the authorsguideline of the Journal.

Abstract:

1.     The abstract is not clear and the objective of the paper is not clearly validated from the abstract.

2.     The future perspective of the experiment should be mentioned in the abstract.

Introduction:

1.      The literature from past work done in the same field missing to strengthen the introduction section. The need and importance of the present experiment should be written in the introduction section.

Materials and Methods:

1. Please try to merge the different sub-sections of the methodology as an individual mention for each component seems a little unscientific method.

2. Introduction, Result, and Discussion sections are poorly cited with the references and required to update and validation with previous studies. The relevant paper listed below may be considered to enhance the scientific quality of manuscript significantly.

·        Kumar, D. et al. (2023). Titanium dioxide nanoparticles potentially regulate the mechanism (s) for photosynthetic attributes, genotoxicity, antioxidants defense machinery, and phytochelatins synthesis in relation to hexavalent chromium toxicity in Helianthus annuus L. Journal of Hazardous Materials454, p.131418. 

Results and Discussion:

1.     The result and discussion section needs to be elaborated more.

2.     The non-significant results was not clearly validated from the previous papers.

3.     Please check and correct the results of the experiments from the tables/figures/graphs provided by authors in the manuscript.

4.     The discussion does not describe the results with proper facts and even does not validate the result with appropriate references.

5.     The discussion did not provide a specific reasons for the results. The provided explanation could have been more satisfactory also.

6.     The strong hypothesis, scientific facts, and validation of previous reports are entirely missing. 

Conclusion:

1.     The conclusion section failed to enlighten the spirit of the finding and is missing the results. Revise it.

2.     In the conclusion section the authors have only mentioned the data but major finding is missing from the conclusion part. Need to rewrite and incorporate this important concern of reviewer.

3.     The conclusion section seems like abstract so there is a need to completely rewrite the conclusion part.

Figures and Tables:

·         Please provide the clear figures and tables.

·         The authors should write the descriptive, elaborated legends for the figures and the tables.

·         Please remove the redundancy from the legends of the figures and tables.

·        The legends of the figures are not crisp and not completely bringing out the sense of the figures. Rewrite it accordingly.

·        The placement of tables and figures in the manuscript should be done appropriately, which is missing in this manuscript. Please revise it.

·        The figures are overlapping the legends, the editing needs to be done.

·        No statistical analysis is mentioned, are results not properly analyzed?

Author Response

Response to comments

Dear editor and reviewers:

Firstly, we would like to express our gratitude for your valuable feedback and acknowledgment of our manuscript, entitled "Analysis of Metabolites and Metabolic Pathways of Three Chinese Jujube Cultivars" (metabolites-2117342), We have diligently reviewed the comments and made the appropriate corrections, which we hope meet your approval. For your convenience in verifying the changes, the revisions are highlighted in red within the revised manuscript. We trust that the modifications are satisfactory and eagerly await your response. The adjustments made based on the reviewers' feedback are detailed below:

Reviewers' comments:

Reviewer #3 

Reviewer’s comments to Authors

  1. Grammatical errors are present, please revise the whole manuscript to remove any possible grammatical and typos errors.

Responses: We have rephrased the whole manuscript and the certificate have uploaded as an attachment. We hope that it now meets with your approval.

  1. Error in sentence formation, please revise the whole manuscript to avoid the use of long sentences.

Responses: We have revised the whole manuscript and the certificate have uploaded as an attachment.

  1. In the keywords, it is strongly advisable to use suitable words that can aid in finding out the manuscript in current registers or indexes. Strictly avoid the use of title words in the keywords.

Responses: Thank you for your good advice. We have revised the keywords.

  1. Please maintain uniformity while in-text citation and referencing.

Responses: Thank you for your good advice. We have checked the whole manuscript and made the change

  1. The reference does not meet the format requirements of the Journal so please check the references as per the authorsguideline of the Journal.

Responses: Thank you for your good advice. We have checked the whole manuscript and made the change

Abstract:

  1. The abstract is not clear and the objective of the paper is not clearly validated from the abstract.

Responses: Thank you for your good advice. We have revised the abstract, and  the objective of the paper have been added in abstract.

  1. The future perspective of the experiment should be mentioned in the abstract.

Responses: The future perspective of the experiment have been added in the end of abstract.

Introduction:

  1. The literature from past work done in the same field missing to strengthen the introduction section. The need and importance of the present experiment should be written in the introduction section.

Responses: The literature from past work have been added in introduction section.

Materials and Methods:

  1. Please try to merge the different sub-sections of the methodology as an individual mention for each component seems a little unscientific method.

Responses: Thank you for your good advice. We have made the change.

  1. Introduction, Result, and Discussion sections are poorly cited with the references and required to update and validation with previous studies. The relevant paper listed below may be considered to enhance the scientific quality of manuscript significantly.

  • Kumar, D. et al. (2023). Titanium dioxide nanoparticles potentially regulate the mechanism (s) for photosynthetic attributes, genotoxicity, antioxidants defense machinery, and phytochelatins synthesis in relation to hexavalent chromium toxicity in Helianthus annuus L. Journal of Hazardous Materials, 454, p.131418.

Responses: We have cited the references

Results and Discussion:

  1. The result and discussion section needs to be elaborated more.

Responses: We have elaborated this section of the manuscript and hope that it now meets with your approval.

  1. The non-significant results was not clearly validated from the previous papers.

Responses: The non-significant results from previous papers were added in result and discussion section.

  1. Please check and correct the results of the experiments from the tables/figures/graphs provided by authors in the manuscript.

Responses: We have check the results of the experiments from the tables/figures/graphs provided by authors in the manuscript and hope that it now meets with your approval.

  1. The discussion does not describe the results with proper facts and even does not validate the result with appropriate references.

Responses: we have validated the result with appropriate references and hope that it now meets with your approval.

  1. The discussion did not provide a specific reasons for the results. The provided explanation could have been more satisfactory also.

Responses: We have made the change and hope that it now meets with your approval.

  1. The strong hypothesis, scientific facts, and validation of previous reports are entirely missing.

Responses: We have made the change.

Conclusion:

  1. The conclusion section failed to enlighten the spirit of the finding and is missing the results. Revise it.

Responses: we have added  the spirit of the finding and results.

  1. In the conclusion section the authors have only mentioned the data but major finding is missing from the conclusion part. Need to rewrite and incorporate this important concern of reviewer.

Responses: Thank you for your good advice. We have made the change.

  1. The conclusion section seems like abstract so there is a need to completely rewrite the conclusion part.

Responses: Thank you for your good advice. We have made the change.

Figures and Tables:

  1. Please provide the clear figures and tables.

Responses: Thank you for your good advice. We will upload  the clear figures and tables as attachment.

  1. The authors should write the descriptive, elaborated legends for the figures and the tables.

Responses: Thank you for your good advice. We have checked the legends for the figures and the tables and made the change.

  1. Please remove the redundancy from the legends of the figures and tables.

Responses: Thank you for your good advice. We have made the change.

  1. The legends of the figures are not crisp and not completely bringing out the sense of the figures. Rewrite it accordingly.

Responses: Thank you for your good advice. We have made the change.

  1. The placement of tables and figures in the manuscript should be done appropriately, which is missing in this manuscript. Please revise it.

Responses: Thank you for your good advice. We have made the change.

  1. The figures are overlapping the legends, the editing needs to be done.

Responses: Thank you for your good advice. We have made the change.

  1. No statistical analysis is mentioned, are results not properly analyzed?

Responses: Thank you for your good advice. We have added the Data analysis in section 2.5

This manuscript is a resubmission of an earlier submission. The following is a list of the peer review reports and author responses from that submission.

Round 1

Reviewer 1 Report

The present research work titled “Analysis of Metabolites and Metabolic Pathways of Three Chinese Jujube Cultivars” provides useful information about the metabolites of the three types of plant studied. The data improved the knowledge about the chemical compounds of jujube. I have some considerations that are described below to improve the quality of the manuscript.

1.      This manuscript descripted many similarities and differences among the three types of jujube, but could be important to give more information about, why, for example, the LZ has lower acids compared with the others.

2.      The introduction should clearly provide the motivation of the comparison of the three types of jujube. The most recent literatures related to metabolites of jujube are also lacking in the introduction.

3.      This is a targeted metabolomic study or an untargeted study? In multiple places the authors indicated this study was “targeted” approach, if so, the authors should provide the list of the targeted compounds in detail.

4.      Line 525-527, the KEGG pathway enrichment analysis was based on the comparison of selected metabolites, how can it cycle back to confirm the accumulation of some certain metabolites.

5.      In conclusion, the analysis of the differences among the three plants was not supported by PCA plots.

6.      The English language should be extensively revised.

Reviewer 2 Report

In this manuscript, the metabolic components of the fruits of Linyi Lizao (LZ), Jiaocheng Sour and Sweet Jujube (STZ) and Xianxian Muzao (MZ) were analyzed. Combined with multivariate statistical analysis, the metabolic components of these three jujube fruits were evaluated and compared by ultra-high performance liquid chromatography-mass spectrometry (UPLC-MS/MS). However, the data form and analysis results of the manuscript are poor, and there is not enough data mining and in-depth analysis. And a lack of innovation to attract more potential readers.

There are a few more issues: Figures 1 and 2 show a lot of information that is not important enough to be included in the main text.

The content shown in Figure 3 is superficial and not highly relevant to the topic.

In addition, the expression of analysis results in Result 3.4 is very confusing and lacks logic.

There are many language problems in the manuscript, so the manuscript version needs to be completely revised to meet the requirements of journal publication.